# Field-Pulse-Induced Annealing of 2D Colloidal Polycrystals

**DOI:** 10.3390/nano13030397

**Published:** 2023-01-18

**Authors:** José Martín-Roca, Manuel Horcajo-Fernández, Chantal Valeriani, Francisco Gámez, Fernando Martínez-Pedrero

**Affiliations:** 1Departamento de Estructura de la Materia, Física Térmica y Electrónica, Universidad Complutense de Madrid, 28040 Madrid, Spain; 2GISC-Grupo Interdisciplinar de Sistemas Complejos, 28040 Madrid, Spain; 3Departamento de Química-Física, Universidad Complutense de Madrid, 28040 Madrid, Spain

**Keywords:** colloidal annealing, 2D confined systems, dynamic self-assembly, fluid interface, superparamagnetic particles

## Abstract

Two-dimensional colloidal crystals are of considerable fundamental and practical importance. However, their quality is often low due to the widespread presence of domain walls and defects. In this work, we explored the annealing process undergone by monolayers of superparamagnetic colloids adsorbed onto fluid interfaces in the presence of magnetic field pulses. These systems present the extraordinary peculiarity that both the extent and the character of interparticle interactions can be adjusted at will by simply varying the strength and orientation of the applied field so that the application of field pulses results in a sudden input of energy. Specifically, we have studied the effect of polycrystal size, pulse duration, slope and frequency on the efficiency of the annealing process and found that (i) this strategy is only effective when the polycrystal consists of less than approximately 10 domains; (ii) that the pulse duration should be of the order of magnitude of the time required for the outer particles to travel one diameter during the heating step; (iii) that the quality of larger polycrystals can be slightly improved by applying tilted pulses. The experimental results were corroborated by Brownian dynamics simulations.

## 1. Introduction

Well-controlled growth of two- and three-dimensional colloidal patterns is essential in the development of photonic crystals, with different applications in electronics, sensors and microlenses [1]. Using an external force to trigger and drive the crystallization process, as well as to promote annealing and tempering, seems to be a logical route toward enhancing the quality of microparticle-based nanocrystals [1,2,3]. In materials science, annealing involves a heating–cooling cycle in which heat rates might be slow and carefully controlled, whereas quenching entails heating the sample to a precise temperature below the critical point. In this work, the structures under study are particle clusters trapped at the interface formed by superparamagnetic particles due to the application of a rotating magnetic field in the plane of the interface [4], while a crystalline order enhancement occurs after the application of out-of-plane field pulses, which, analogous to what occurs at the atomic level with increasing temperature, can promote the breaking of interparticle bonds and an increase in the particle mobility. The objective of the study is to explore the parameter space of the field able to promote order enhancement in colloidal planar polycrystals.

Thermal Brownian micro- or nanoparticles with tunable size, shape and interactions have been used as model systems to understand the physical mechanisms controlling static and dynamic self-assembly, the formation of disordered aggregates and gels or to corroborate the physics underlying different phase transitions [5,6]. In processes leading to the formation of equilibrium phases and static self-assemblies, the system tends to reversibly reach local or global minima in the free energy landscape, and consequently, the pattern of the equilibrium configuration depends on the interparticle interactions. Although the equilibrium states can be reasonably predicted by thermodynamics, a proper explanation for the kinetics of the process remains challenging [6]. On the other hand, in dynamic self-assemblies, the promoted structures are continuously dissipating energy, which plays a relevant role in the main features of the out-of-equilibrium structure [7]. Understanding the different mechanisms governing these processes is of practical importance. For instance, controlling the growth of two- and three-dimensional colloidal patterns is essential for the design, synthesis and development of smart materials [8], such as photonic crystals [9,10,11,12], micro-transporters [13] or data storage devices [14].

In colloidal materials formed with self-assembly methods, microparticles, unlike nanoparticles, tend to form low-quality crystals comprised of different domains of hundreds of particles oriented along random directions, in which crystal-void interfaces and defects are widespread [8,15]. To have control over the structural order and the averaged size of the crystal grains, which is mandatory in the manufacture of materials with improved physical properties, it is common to use annealing strategies based on the controlled application of external stimuli that promote both the melting of defective areas and the subsequent recrystallization [16,17,18,19,20]. In atomic and molecular engineering, where it was first defined, annealing is a process of adjusting the grain boundary density of a polycrystalline material, such as a metal, ceramic material, rocks, proteins or ice, by heating and then cooling at a controlled rate [21,22]. This process, essential in macromolecular, biotechnology and a variety of metallurgical, geological and meteorological phenomena, has a major influence on material properties such as yield strength or electrical conductivity, both in 3D and 2D [23,24]. At the microscale, annealing processes can be tracked directly by optical microscopy [25] and promoted by the use of holographic optical tweezers [19], crystal agitation [18], the inclusion of active particles [2,26] or the use of tunable colloids that are periodically exposed to different conditions, in the vicinity of their melting points [17].

In this context, the interactions between colloidal particles, and thus the melting and crystallization processes, can be regulated in real-time by applying externally generated fields [1,3]. For example, Bevan and co-workers designed a closed-loop control scheme that allows switching from polycrystalline states to a single-domain crystal through the monitored application of an electric field [27,28]. Alternatively, different authors have used strategies similar to the heat treatment of metals, in which external fields were introduced cyclically. By alternating the external field, the systems switched intermittently from high-temperature states, where small fields induced weak attractions, to low-temperature states, where strong fields induced strong attractions, so kinetically arrested suspensions were allowed to periodically relax and find lower energy states through local rearrangements of the suspension structure [29,30,31]. Kao et al. found that if the disorder is produced by thermal diffusion, the optimal cyclic conditions arise when the deactivation duration is about half the characteristic melting time of the system [32].

The present work focuses on the development, study and optimization of an adjustable annealing mechanism applied to finite 2D colloidal polycrystals. In finite 2D polycrystals formed by attractive colloids, premelting occurs even before the melting point is reached when the value of the chemical potentials of the liquid and solid phases coincide [33]. This process is favored at the contour and at defects, as these zones have higher free energy than the internal areas of the crystal [34]. Pre-melting can occur as a complete process, in which the crystal surface melts into a liquid layer that increases in size as it approaches the melting point, or as an incomplete premelting, in which the liquid layer increases as it approaches the melting point but remains finite [6]. In finite 2D crystals with thermal premelting, first-order melting occurs from both edges and within the bulk through a grain-boundary mediated mechanism [35]. In freezing, the presence of defects plays a minor role, and the increase in the particle attractions causes larger chemical potential differences and a faster crystallization rate [6]. Consequently, in finite 2D colloidal crystals, annealing is affected by multiple factors, such as crystal size, heating and cooling rates, the presence of defects, inter-particle interactions or the application of external fields. The effects of these issues on the annealing mechanism are still poorly understood, especially in colloidal systems, where they have not yet been studied in depth, even though colloidal interactions are much simpler than atomic ones. As a consequence, these systems remain the simplest model for the study of annealing mechanisms.

In this study, we use magnetic colloids adsorbed on a fluid interface as a model system to study annealing cycles in 2D. The use of magnetic colloids in the study of the annealing of 2D colloidal polycrystals is justified since they allow the induction of interparticle interactions with different characteristics—range, anisotropy and intensity—easily tunable by the action of an externally applied magnetic field of moderate intensity. In the explored configuration, in which the particles are forced to reside in the plane of the interface, the particles have easily tunable attractive and repulsive interactions, isotropic or anisotropic, through modulation of the external magnetic field so they can be assembled or disassembled in a controlled manner [4,36,37]. Concretely, the adjustment of the angle between the confining interface and the precessing applied field, which here plays the role of temperature in molecular systems, allows for accelerating or decelerating both the melting and the freezing rates [38,39,40]. Next, we study in detail the effect of polycrystal size, pulse duration, pulse slope and pulse frequency on the efficiency of the annealing process. The applied methodology has an important advantage over other annealing strategies used in colloids, where the melting of metastable defective states is driven by thermal energy in a temperature range that may not differ much from room temperature [27,28,32,35], thus extending the capabilities of colloidal crystals to understand solid-state phenomena under thermal annealing.

## 2. Materials and Methods

### 2.1. Magnetic Colloids

The colloidal suspension consists of superparamagnetic beads coated with polymer ending with a carboxylic acid group immersed in deionized, ultrapure water (Dynabeads^®^ M270, supplied by Invitrogen). The particles consist of a highly cross-linked polystyrene matrix in which superparamagnetic grains of γ-Fe_2_O_3_ and Fe_3_O_4_ are uniformly dispersed. The microspheres are relatively monodispersed, with a radius of *a* = 1.4 µm and retain the superparamagnetic character of iron oxide grains. The magnetic field required to saturate the magnetization of the particles is of the order 100 kA/m, and the particle susceptibility χ = 0.4 [41]. In addition, they exhibit excellent dispersibility and easy handling in a variety of aqueous media. The presence of the surface carboxyl groups confers a negative charge to the microparticles, and the screened electrostatic repulsion prevents particles from irreversible coagulation at a minimum primary energy. To eliminate possible residues, the dispersions were washed three times in a 5.2 mM solution of SDS (sodium dodecyl sulfate) before the experiments. During the first step of this process, the particles were attracted to the bottom of an Eppendorf with the help of a neodymium magnet. In successive steps, the system was kept at rest for a couple of hours to allow the particles to precipitate gently, minimizing the formation of permanent aggregates. At the end of each of these steps, the aqueous medium was replaced with deionized water and the mixture was homogenized in an ultrasonic bath to break up any aggregates that may have formed during washing.

### 2.2. Adsorption at the Fluid Water/Decane Interface

To attract the particles to the water/decane interface and facilitate the adsorption process, a neodymium magnet is used. The decane was previously passed through an alumina column three times to remove any contaminants. The adsorption of the particles at the interface can be facilitated by using salts or cationic surfactants, but we have chosen not to add them, as it has been observed that they not only favor adsorption but also the formation of permanent aggregates, which hinder crystallization [42]. Once adsorbed on an oil/water fluid interface, the magnetic particles are preferentially submerged in the aqueous phase due to their hydrophilic character, which accounts for a small value of the contact angle and a small influence of capillary and electric dipole interactions [39]. Hence, the structures formed are determined almost exclusively by the character and strength of the imposed magnetic interactions. During the experiments, the laden water/decane interface is confined by a hollow, non-magnetic glass cylinder to reduce the drift motion generated by convention effects, which are difficult to eradicate in such systems. After preparing the particle-loaded fluid interface, the non-adsorbed particles fall into the aqueous sub-phase as soon as the magnet is removed, while the adsorbed particles remain in the plane of the fluid interfaces due to the relatively high value of the trapping energy.

### 2.3. Field Generation and System Monitorization

The application of a magnetic field allows the colloidal particles to be externally energized. Current-carrying coils oriented orthogonally to each other are used to generate constant, rotating or precessing fields. The fields generated by this coil configuration can reach 10 mT and are uniform in the field of view of the microscope, which minimizes the generation of magnetic forces and avoids the occurrence of unwanted collective movements. The formation of magnetic colloidal crystals adsorbed at the fluid interface together with the file-induced annealing process was visualized in real-time by bright field optical microscopy. Particularly, an Olympus BH2 optical microscope with a 20 × 0.25 NA objective and a working distance of approximately 1 cm connected to an Edmund EO1312M CCD camera, was employed. The laden and confined fluid interface was placed on the stage of this microscope, in the center of the coil assembly, and the magnetic fields were applied for the magnetization of the adsorbed colloidal particles. VirtualDub video editing software was used to process the recorded videos, and ImageJ software was used to determine the central position of the particles. The fact that colloids and domain walls can be visualized directly using a bright field microscope makes them ideal candidates for modeling melting and crystallization processes. The adsorption method and the experimental setup are shown in Figure 1a. In the experiments, the temperature was approximately 298 K.

### 2.4. Formation of the Polycrystals

In the first stage of this study, a rotating field is applied in the horizontal plane from the combination of two sinusoidal components, 90° out of phase, oriented along the X and Y axes, amplitude $H_{x}$ and $H_{y}$, and with identical angular frequency $\omega_{x}=2\pi f_{x}=\omega_{y}=2\pi f_{y}$, characterized by the field strength $H_{0}=\sqrt{H_{y}^{2}+H_{x}^{2}}$. At low frequencies, this time-dependent field forces the formation of elongated structures that vibrate or rotate, synchronously or asynchronously, with the applied field [43]. If the frequency of the field is higher than 10 Hz, however, an averaged attractive potential promotes the formation of circular assemblies or chains composed of rotating particles, which, in turn, rotate or vibrate in the plane of the field [37,44]. When $H_{x}$ and $H_{y}$ are similar, the application of the high-frequency rotating field induces an isotropic effective attractive dipolar potential at the interface plane, $\langle U_{dd}\rangle =-\left[\mu_{0}m^{2}/8\pi {\left(x^{2}+y^{2}\right)}^{3/2}\right]$, where $\mu_{0}$ is the vacuum magnetic permeability and m is the induced magnetic moment in the particles, and promotes the formation of two-dimensional colloidal polycrystals. In these self-assembled polycrystals, crystalline domains with hexagonal order and composed of tens of particles are randomly oriented. When the local monomer density is adequate, the crystallization process occurs in the range of tens of seconds, and the resulting polycrystals are well separated and dispersed across the fluid interface (Figure 1b shows a sequence of images of the polycrystal formation). Once formed, the rotating field is replaced by a time-dependent field resulting from the combination of two oscillating perpendicular fields applied in the plane of the interface with different frequencies (Figure 1b, t = 120 s). The latter still promotes isotropic attraction between the particles but prevents rotation of both the constituent particles and the resulting polycrystals, which could have some effect on the annealing process. In Figure 1b, we show the evolution of the polycrystal orientation in the 90–95 s range with that in the 120–130 s range. The stability of the colloid structures resulting from the above methodology has been characterized in Figure 1c, where the promotion of these circular symmetrical structures is observed in the white area, while in the blue and green areas, the particles tend to form linear or partially disordered structures. The described strategy is repeated in each experiment after melting any eventual structure previously formed, so that a polycrystalline configuration is always used as a starting point in each annealing process.

### 2.5. Pulse Effect

To induce the restructuring of the formed 2D polycrystals and to explore the possibility of improving their spatial and orientational order, a series of magnetic field pulses are applied to the system. The series of magnetic pulses, applied outside the plane interface, is given by the following expression:(1)$$\begin{array}{c} H_{pulso}=H_{z}^{\max}\cdot \frac{1}{4}\left\{1-\frac{\left|\sin\left(\omega_{pulse}t\right)\right|}{\sin\left(\omega_{pulse}t\;\right)}\right\}\cdot \left\{\;1+\frac{\left|\sin\left[\omega_{pulse}\left(t-\tau_{p}\right)\right]\right|}{\sin\left[\omega_{pulse}\left(t-\tau_{p}\right)\right]}\;\right\}\hat{z}+\\ +H_{x}^{\max}\cdot \frac{1}{4}\left\{1-\frac{\left|\sin\left(\omega_{pulse}t\right)\right|}{\sin\left(\omega_{pulse}t\;\right)}\right\}\cdot \left\{\;1+\frac{\left|\sin\left[\omega_{pulse}\left(t-\tau_{p}\right)\right]\right|}{\sin\left[\omega_{pulse}\left(t-\tau_{p}\right)\right]}\;\right\}\hat{x}\; \end{array}$$
where $H_{z}^{max}$ and $H_{x}^{max}$ are the maximum values of the field components along the Z and X axes, respectively. This value is given by the sum of the amplitude of the square wave $A_{i}$ and the field offset $H_{i}^{0}$, $H_{i}^{max}=A_{i}+H_{i}^{0}\;$(i = z, x). In Equation (1), $\tau_{p}$ is the pulse duration and $\omega_{pulse}=\frac{2\pi }{T}=2\pi f_{pulse}$ the angular frequency of the square wave. The inter-pulse relaxation time is defined as $\tau_{r}=T-\tau_{p}$, where *T* stands for the wave period (Figure 2a). The combined application of the time-dependent field in the interface plane and the out-of-plane square pulses causes the induced moments on the particles to tilt outward from the interface. As the particles are strongly confined in the plane of the interface, the reorientation of the magnetic moments does not cause the orientation of the structures formed outside the interface but modulates the angle γ between the induced moments and the line connecting the particles, and thus the intensity and character of the anisotropic magnetic dipole interaction. When both fields are applied simultaneously, during the pulse duration, the attraction induced by the in-plane component of the field, responsible for holding the crystal together, is partially balanced by the repulsion promoted by the out-of-plane field, which can eventually cause the partial disintegration of the structure and allows the reconstruction of the colloidal crystal. In the dipole–dipole approximation, the time-averaged potential of two paramagnetic spheres adsorbed on a planar fluid interface when under the action of a field in precession about the Z axis can be expressed as the sum of the in- and out-of-plane contributions, $\langle U_{dd}\rangle =\frac{\mu_{0}\chi^{2}}{2\pi r^{3}}\left({\left(H_{z}^{max}\right)}^{2}/2-H_{0}^{2}\right)$ [37]. Hence, if we define the tilt angle of the field with respect to the interphase as $\gamma =\mathrm{atan}\left(\frac{H_{z}^{\max}}{H_{0}}\right)$, and the effect of mutual induction is ignored, the critical angle that separates the attraction and repulsion between two adsorbed particles is $\alpha_{c}=54.7{}^{\circ}$ [44]. We have corroborated that for smaller angles there is no change of order due to the application of the pulse while above this value, the repulsion induced by the pulse is comparable to or greater than the attraction due to the in-plane field.

### 2.6. Degree of Alteration of the Order during the Annealing Process

The change of the hexagonal order is quantitatively studied through the 6-fold bond-oriental order function $g_{6}\left(r,\;t\right)$, where r is the radial distance. For a site k whose nearest neighbor interparticle vectors are labeled by j form angles $\theta_{kj}$ from a given reference direction, this order function is expressed as follows:(2)$$g_{6}\left(r\right)=\frac{\langle \varphi_{6,k}^{*}\left(0\right)\cdot \Psi_{6,k}\left(r\right)\rangle }{\rho \left(r\right)}=\frac{1}{N}\left|\frac{\sum_{k=1}^{N}\varphi_{6,k}^{*}\left(0\right)\cdot \Psi_{6,k}\left(r\right)}{\rho_{k}\left(r\right)}\right|$$

Here, $\varphi_{6,k}^{*}\left(0\right)=\frac{1}{6}\sum_{j=1}^{6}e^{-i6\theta_{kj}}$ is the complex conjugate of the local order parameter of particle *k*, $\Psi_{6,k}\left(r\right)=\sum_{k=1}^{N}\delta \left(r-r_{k}\right)\varphi_{6,k}$ is the angular average of the orientational order density parameter, N is the number of particles and $\rho \left(r\right)=\sum_{k=1}^{N}\delta \left(r-r_{k}\right)$ refers to the microscopic particle density. In a monolayer of particles exhibiting low hexagonal order, $g_{6}\left(r\right)$ is expected to exhibit a value close to zero. In large crystals, $g_{6}\left(r\right)$ is expected to show peaks with values close to 1 and remain constant over a significant range of distances. In small crystals and polycrystals, a rapid decrease in the function is expected. The temporal orientational function, $\varphi_{6}\left(t\right)$, measures the time evolutions of the orientational order and is defined as follows:(3)$$\varphi_{6}\left(t\right)=\frac{1}{2N^{2}}\left|\overset{N}{\underset{k=1}{\sum }}\overset{N}{\underset{j=1}{\sum }}e^{-i6\theta_{kj}\left(t\right)}\right|$$

To describe the time evolution of the degree of order of the crystals we have followed different methods. In the first one, a linear curve has been fitted to the first 7 maxima of $g_{6}\left(r,t\right)$ before any pulse is applied to the system, $f_{0}\left(r,t_{0}\right)=a_{0}\left(t_{0}\right)\cdot r+b_{0}\left(t_{0}\right)$, and just before each new pulse *i*, $f_{i}\left(r,t\right)=a_{i}\left(t\right)\cdot r+b_{i}\left(t\right)$, see Figure 2b, so that the dimensionless parameter $\xi_{i}$ is defined as follows:(4)$$\xi_{i}\equiv \frac{a_{i}-a_{0}}{\left|a_{0}\right|}=\frac{a_{i}}{\left|a_{0}\right|}-1\;$$
and $\xi \equiv \frac{\langle a_{i}\rangle {}_{steady}-a_{0}}{\left|a_{0}\right|}$ represents the average taken over the last steady cycles, when the polycrystal no longer evolves appreciably after the application of new pulses. Here, it is important to stress that ξ, which describes the degree of ordering in each individual polycrystal, is normalized by $a_{0}$, which is different for each specific initial configuration. The function is positive when an improvement in order is observed $\left(\langle a_{i}\rangle {}_{steady}>a_{0}\right)$, negative when the crystalline order degrades $\left(\langle a_{i}\rangle {}_{steady}<a_{0}\right)\;$and approximately zero when the degree of order does not change significantly $\left(\langle a_{i}\rangle {}_{steady}\approx a_{0}\right)$. The two next approaches are only applied to polycrystals composed of a relatively low number of domains, typically less than 10. In the first one, we measure the area $S_{i}$ of the largest region composed of connected particles having $\varphi_{6,k}>0.8$ (Figure 2c), just before each pulse *i* + 1, when the polycrystal has reached a steady state. Hence, we assess the time evolution of the area fraction, defined as the change in the area of the dominant domain divided by the total area of the polycrystal (Figure 2d). The change in the surface of the predominant domain is also measured in relation to its initial area, $S_{0}$, as $\alpha =\frac{\langle S_{i}\rangle -S_{0}}{S_{0}}$, where 〈…〉 represents the average is taken over the last steady cycles. In the second method, we follow the time evolution of the parameter $\varepsilon \equiv \frac{\langle \varphi_{6}{{}_{,min,i}\rangle }_{steady}-\varphi_{6}{}_{,min,1}}{\varphi_{6}{}_{,min,1}}$. Here, $\varphi_{6}{}_{,min,1}$ is the minimum value adopted by $\varphi_{6}\left(t\right)$ during the application of the first pulse, and $\langle \varphi_{6,min,i}\rangle_{steady}$the average taken over the last steady cycles (Figure 2d). This minimum value taken by $\varphi_{6}\left(t\right)$ during the application of each pulse is related to the area of the seeds, i.e., those crystalline zones that remain roughly unaltered during the application of the pulses. In summary, Figure 2d shows the time evolution of $\varphi_{6}\left(t\right)$, $\xi_{i}$ and the fraction of area covered by the predominant domain, all magnitudes required to calculate the different order parameters ξ, α and ε.

### 2.7. Simulations

The annealing process observed in the two-dimensional magnetic colloidal system under the magnetic pulse provided by Equation (1) was assessed with Brownian dynamics (BD) simulations. In BD simulations, the effect of the solvent on the trajectories is approximately incorporated by means of a random contribution, in the framework of the Langevin equation [45,46]. The particle position $r_{i}\left(t+\Delta t\right)$ is expressed as follows:(5)$$r_{i}\left(t+\Delta t\right)=r_{i}\left(t\right)+\frac{D}{kT}\underset{j\neq }{\sum }F_{ij}\Delta t+\xi {\left(2D\Delta t\right)}^{1/2}\hat{w},\;$$
where $\xi \hat{w}$ is a stochastic Gaussian vector of zero mean and unitary variance. *D* stands for the (short-time) diffusion coefficient of the particles, experimentally determined from the mean squared displacement under field-free conditions, and *k* is the Boltzmann constant and T is the absolute temperature. The force acting on particle *i* mediated by particle j is denoted as $F_{ij}$. This term comprises both, the excluded volume interaction and the magnetic contribution.

The excluded volume term was taken into account in the hard sphere framework, and particles overlap was avoided by a modification of the method proposed by Schaertl and Sillescu [47] as follows. In a time step, all particles move simultaneously according to Equation (5). If overlapping occurs between two particles, they are separated in the direction of their relative position vector. If the particles continue to overlap after this step, the initial position of the particles is restored.

The magnetic force contains an effective In-plane attractive force and an out-of-plane repulsive pulsating force, induced by the magnetic field expressed in Equation (1). In the dipolar approximation [48], the force between the magnetic dipole moment of particles *i* (*m_i_*) and *j* (*m_j_*), separated by a distance *r_ij_*, can be expressed as follows:(6)$$F_{ij}=\frac{3\mu_{r}\mu_{0}}{4\pi r_{ij}^{4}}m^{2}\left[\left({\hat{m}}_{i}\cdot {\hat{r}}_{ij}\right){\hat{m}}_{j}+\left({\hat{m}}_{j}\cdot {\hat{r}}_{ij}\right){\hat{m}}_{i}+\left({\hat{m}}_{i}\cdot {\hat{m}}_{j}\right){\hat{r}}_{ij}-5\left({\hat{m}}_{i}\cdot {\hat{r}}_{ij}\right)\left({\hat{m}}_{j}\cdot {\hat{r}}_{ij}\right){\hat{r}}_{ij}\right]$$
where $\mu_{r}$ is the relative magnetic permeability. The values of the magnetic dipole moments are evaluated on-the-fly in each configuration under the condition proviso *H_total_* >> *H_dip_*. In this regime, the following equations hold:(7)$$m_{i}=\left(1-\frac{\chi }{24}\underset{j\neq i}{\sum }\frac{1}{r_{ij}^{3}}\right){\hat{H}}_{\mathrm{total}}^{\mathrm{ext}}+\frac{\chi }{24}\underset{j\neq i}{\sum }\frac{{\hat{H}}_{\mathrm{total}}^{\mathrm{ext}}\cdot r_{ij})}{r_{ij}^{5}}{\hat{r}}_{ij},$$
where $H_{total}^{ext}$ represents the total external magnetic field applied on the system and χ is the magnetic susceptibility under static field. Finally, dimensionless variables are defined as follows: (i) lengths are reduced using the experimental particle diameter *σ*, (ii) times are reduced with *τ* = *σ*^2^*/D*, (iii) magnetic field intensities are divided by *μ*_0_*H*_0_ and, finally, (iv) energies are reduced with the thermal energy kT. Because of the very strong magnetic interaction induced by the pulse magnetic field in the Z-direction, the integration time were *Δt/τ* = 10^−9^. The simulations were run in a squared box and seeded with the experimental coordinates of a medium-size cluster (*N* ≈ 330 particles) obtained for values of *μ*_0_*H_total_* = 5.5 mT, *τ_r_/τ* = 6 × 10^8^ and *τ_p_* = 0.15 s. As usual, during the analysis of the trajectories, the drift of the center of mass of the simulation box is subtracted during the dynamics [49].

## 3. Results

Throughout the remainder of this paper, we will explore how the application of the pulses can alter the order of the formed 2D colloidal polycrystals. Considering that in all the performed experiments $f_{x}=20\;\mathrm{Hz}$, $f_{y}=60\;\mathrm{Hz}$, $A_{i}=H_{i}^{0}$ and $\gamma =57^{o},$ the parameter space to explore is$\;H_{0},\;H_{z}^{max},\mathrm{the}\;\mathrm{pulse}\;\mathrm{tilt}\;\mathrm{angle},\;\beta =\mathrm{atan}\;\left(\frac{H_{z}^{max}}{H_{x}^{max}}\right),f_{pulse},\;\mathrm{and}\;\tau_{p}$.

### 3.1. Effect of the Pulse Duration $\left(\tau_{p}\right)$

In the next series of experiments, the effect of the duration of the periodical application of a pulse of vertical fields on the final order of the colloidal polycrystals is studied. Here, a Z-axis square field is applied, keeping constant the set of fixed parameters, β = $90^{o}$, $\mu_{0}H_{x}^{max}\;$= 0.0 mT, $\mu_{0}H_{z}^{max}\;$= 2.0 mT (5.5 mT or 8.0 mT), while varying the pulse duration. It is important to note that in each experiment, the maximum value of the pulse frequency is strongly determined by the chosen pulse time, the applied field strength, or the size of the polycrystal since when pulses are too frequent, polycrystals cannot be completely reformed before the next pulse occurs. To ensure stabilization of the system after each pulse, the period of the perpendicular actuation used in the experiments was varied when increasing the pulse duration, between 10 and 20 s, for $\mu_{0}H_{z}^{max}\;$= 2.0 mT, or between 5 and 10 s for $\mu_{0}H_{z}^{max}=8.0\;\mathrm{mT}$.

#### 3.1.1. Small Polycrystals

First, we analyze small polycrystals, consisting of between 100 and 500 particles, which after being subjected to the protocol described in Section 2.4 form less than 10 domains (Figure 3a). As mentioned above, the experiments were performed on different initial polycrystalline configurations. Figure 3b shows the dependence of the order parameters *ξ*, *α* and *ε* on the pulse duration $\tau_{p}$. Short pulses, $\tau_{p}<0.1\tau_{c},$ where $\tau_{c}$ is defined as the average time it takes for the outer particles to travel a distance equal to the particle radius when subjected to only one external pulse, do not perturb the system sufficiently to promote any improvement in the order of the assembled structures, and the values of *ξ*, *α* and *ε* remain practically zero. In the interval between 0.1 *τ_c_* < *τ_p_* < 2 *τ_c_*, the pulses are capable of inducing partial disordering of the structure. For the chosen set of field parameters, the application of the pulses shows some effectiveness and *ξ, α* and *ε* take positive values. Over this range of crystal sizes, field strengths and pulse durations, first-order fusion emerges preferentially at crystal edges and grain boundaries, expanding over the entire area of the smallest domains [6,16]. At the end of each pulse, the surviving bigger domains serve as crystal seeds for the growth of new domains so that the larger domains grow at the expense of the smaller ones, similar to what happens in ferromagnetic materials when an external field is applied [50]. This mechanism resembles the Ostwald ripening, where small clusters dissolve in favor of larger ones that are energetically more favorable. The mechanism is quite robust so that after a few pulses, most of the particles are part of the predominant domain, and the application of new pulses only induces disorder of the outer particles, which at the end of each new pulse are mostly incorporated into the dominant domain [51] (see Appendix A and Figure 3a). For values of *τ_p_* > 5*τ_c_*, the outer particles travel such a distance that the displacements of the inner particles are allowed, and a highly disordered fluid state is promoted along the entire area of the polycrystals. At the end of each pulse, the in-plane field again promotes freezing, but the fact that in each post-pulse configuration, the assembly is disordered, and the particles are separated causes them to preferentially self-assemble into new polycrystalline configurations, so *ξ*, α and ε adopt zero or negative values. Besides, Figure 3b shows how the trends measured under different field strengths tend to collapse when the pulse duration is normalized by *τ_c_*. Here, *τ_c_* = 0.1, 3.5 and 5 s when ${µ}_{0}H_{total}^{ext}={µ}_{0}\sqrt{H_{0}^{2}+{\left(H_{z}^{max}\right)}^{2}}=$ 2.0 mT, 5.5 mT and 8.0 mT, respectively. As expected, the optimal pulse duration that helps to improve the order in polycrystals decreases with increasing field strength and approaches the time that the outer particles need to travel a radius distance. This above result is corroborated by preliminary Brownian dynamics simulations as presented in Section 3.2.

#### 3.1.2. Large Polycrystals

In large polycrystals comprised of more than 500 particles, the strategy based on the application of magnetic field pulses along the Z direction is less efficient, and we only detect positive values of *ξ* at relatively high field strengths, ${µ}_{0}H_{total}^{ext}={µ}_{0}\sqrt{H_{0}^{2}+{\left(H_{z}^{max}\right)}^{2}}=$ 8.0 mT, when applying slightly longer pulses than those measured on small polycrystals (Figure 4a). In these large polycrystals, the mobility of the inner particles is strongly hindered by the presence of the surface region so that when the pulse duration is short, premelting is only detected in the peripheral zones (see Appendix A). Increasing the pulse duration allows the inner parts to melt while pushing the outer particles away. The result is a disordered solidification of the corona and only an incomplete improvement of the order in the core. To enhance the effect that the pulse can have in increasing the order of large polycrystals, we have applied inclined pulses, resulting from the simultaneous application of pulses along the Z and X axes. The variation of the pulse tilt angle, *β*, affects the symmetry of the interaction between particles during the application of the pulses. Under the action of the resulting elongated precessing field, the attraction becomes stronger, and the structures formed tend to stretch in the favored direction (Figure 4b). In principle, this imposed configuration can improve the order of the crystals, as it favors the growth of domains oriented along the X direction, or it can worsen the order of the crystals by generating gaps and voids in the structure. Figure 4c shows how, as *β* decreases, there is a slight enhancement in the hexagonal order due to the formation of the privileged direction in the crystal.

### 3.2. Simulations Results

Here we present the results for one of the simulations carried out using the method explained in Section 2.7. As can be seen in Figure 5, when applying the pulses of field that promote the precession of the induced dipoles about the perpendicular to the confining flat boundary, a qualitatively similar behavior to that presented by the experiments has been observed, with an improvement in the global hexagonal order, $\varphi_{6}\left(t\right)$, and of the local hexagonal correlation, $g_{6}\left(r,t\right)$. In this experiment, three different pulses of duration $\tau_{p}=0.15\;s$ were applied every 5 s. Consequently, $\varphi_{6}\left(t\right)$ increases from 0.1 to 0.3 while the function $g_{6}\left(r,t\right)$ decays more slowly with distance.

As it clearly appears, simulations can be used not only to corroborate experimental findings by eliminating experimental factors that make characterization difficult, such as the occurrence of permanent aggregates between colloids, and by controlling the initial configuration, size and number of domains of the clusters, but also to study other systems that are difficult to access experimentally, such as mixed systems of two different types of particle sizes with radii smaller than 1 micron, for which the resolution of the microscope does not allow the correct identification of individual particles within the clusters.

## 4. Discussion

Understanding the mechanisms of annealing is fundamental in metallurgy or materials science, where this strategy is used to increase its ductility and reduce its hardness, making it more workable [52,53]. In this work, we introduce a new annealing mechanism for 2D polycrystals in which colloidal monolayers of superparamagnetic particles, adsorbed at a planar water–air interface, are melted and crystallized by the action of a pulsed magnetic field that periodically precesses about the axis perpendicular to the interface. By studying the field-induced annealing process in real space by microscopy and by computer simulations, we have investigated the dependence of different order parameters, which reflect the degree of crystallinity of the studied polycrystals, as a function of their size as well as the main characteristics of the applied pulses (duration time, frequency, intensity, inclination). We have found that the proposed strategy is most effective when the pulse duration is strong and long enough to promote disorder in the smaller, peripheral domains but weak and short enough to allow some domains to survive, which act as seeds for the next crystallization. The annealing of larger polycrystals is more difficult, as the application of intense pulses needed to melt the inner part of the polycrystals also causes excessive corona disorder. In these systems, only the application of inclined pulses, which promote the formation of aligned domains, was able to generate a slight improvement. The results here described can be useful in the construction of photonic crystals—an ordered array of interstitial voids that act as a diffraction grating when the interstitial spacing is similar to the wavelength of the incident light—the synthesis of single-crystalline nanoparticles, to achieve ultranarrow surface lattice resonances [54], or as an ideal basis for focused ion beam milling. In addition, analogous methods could be used in the controlled production of 2D mesocrystals, aggregates of nanocrystals with aligned crystalline axes, when the latter have magnetic properties and relatively large sizes [55]. In the near future, we want to address the study of binary monolayers composed of particles of different sizes or explore the effect of applying more gradual excitations. In this context, the predictive information encompassed by computer simulations seems to be crucial for the following several reasons: (i) the trajectories are free from eventual undesirable but unavoidable experimental events such as the formation of irreversible clusters that may blur the observable results; (ii) it allows to explore conditions that are experimentally demanding. All these features will be exploited in a forthcoming publication.

## Figures and Tables

**Figure 1 nanomaterials-13-00397-f001:**
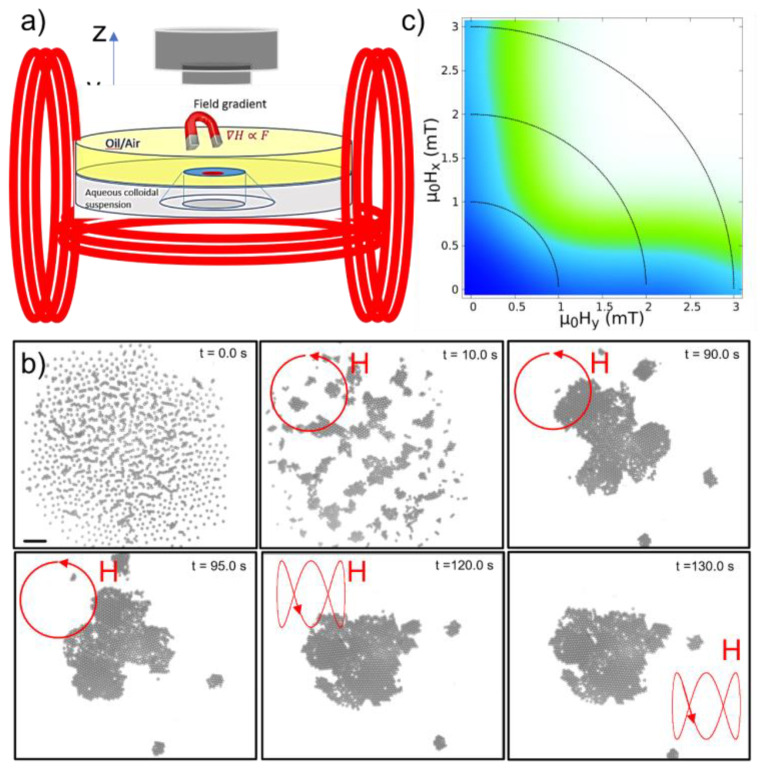
(**a**) Magnetic particles are attracted to a water/decane interface with the help of a magnet to facilitate the adsorption process. After the magnet is moved away, the non-adsorbed particles fall into the aqueous sub-phase, while the adsorbed particles remain in the plane of the fluid interfaces due to the relatively high value of the trapping energy. To reduce the drift motion generated by convention effects, the laden water/decane interface is confined by a hollow, non-magnetic glass cylinder. Finally, the adsorbed magnetic particles are magnetized by the field generated by a pair of coils connected in series, aligned along the X and Y axes and a fifth coil aligned along Z, the optical axis of the microscope. (**b**) The image sequence shows how the application of the high-frequency rotating field, $f_{x}=f_{y}=20\;\mathrm{Hz}$, at the interface plane promotes a crystallization process, which, at relatively high particle densities, occurs in the range of tens of seconds. From t = 120 s, the rotating field is replaced by the combination of two fields with different frequencies, which still promotes isotropic attraction between the particles but prevents rotation of both the constituent particles and the resulting polycrystals (please, compare the images taken at t = 120 s and 130 s). Scale bar: 20 microns. (**c**) Diagram of configurations showing the regions where crystal formation is observed after application of the in-plane rotating field. The white area represents the conditions where the attraction between colloids causes the particles to form a polycrystal composed of different domains with hexagonal order. In the blue zone, the colloidal particles form linear aggregates or are scattered by the thermal noise itself, and the green zone represents the onset of the premelting zone, a transition zone between the two previous configurations. Here, the continuous lines represent constant field values.

**Figure 2 nanomaterials-13-00397-f002:**
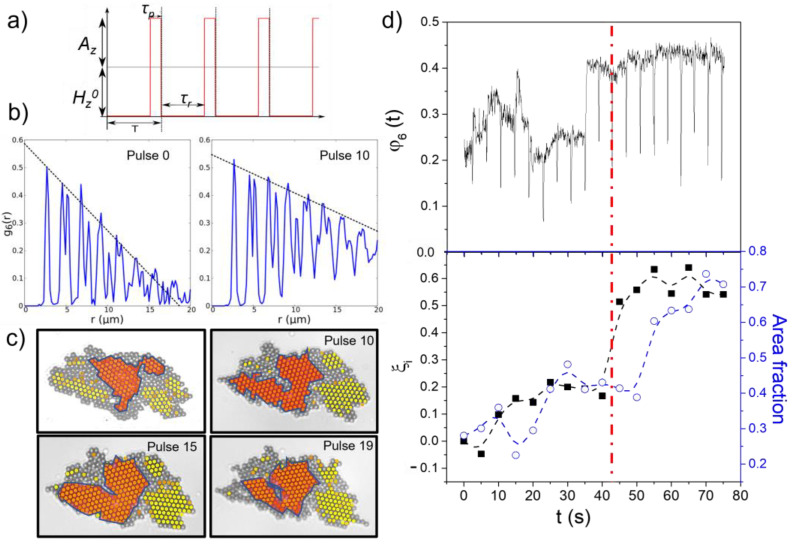
(**a**) Series of pulses are applied outside the planar interface to induce the restructuring of the formed 2D polycrystals and to explore the possibility of improving their spatial and orientational order. The pulses, applied perpendicular to the fluid interface, are determined by the square wave amplitude $A_{z}$, the field offset $H_{z}^{0}$, the pulse duration$\;\tau_{p}$ and the period of the square wave *T*. (**b**) In the first method, the degree of alteration of the hexagonal order during the annealing process is evaluated by following the change in the slope of the linear fit at the first 7 maxima of $g_{6}\left(r,t\right)$ after each pulse. (**c**) In an alternative strategy, the degree of alteration of the hexagonal order during the annealing process is evaluated by following the change in the area of the predominant crystalline domain after each pulse, defining this area as the surface covered by the largest region (area colored in orange) composed of connected particles having $\varphi_{6,k}>0.8$ (particles colored in yellow). (**d**) The upper plot shows the time evolution of $\varphi_{6}\left(t\right)$, while the lower graph shows the time evolution of both, $\xi_{i}$ and the fraction of area covered by the predominant domain. Here, $\xi_{i}$ is the relative change in the slope of the line fitted to the first maxima of $g_{6}\left(r,t\right)$ after each pulse *i*. The measured data, corresponding to the polycrystal presented in c, show that the system reaches a stable conformation after 10 pulses (red dashed line).

**Figure 3 nanomaterials-13-00397-f003:**
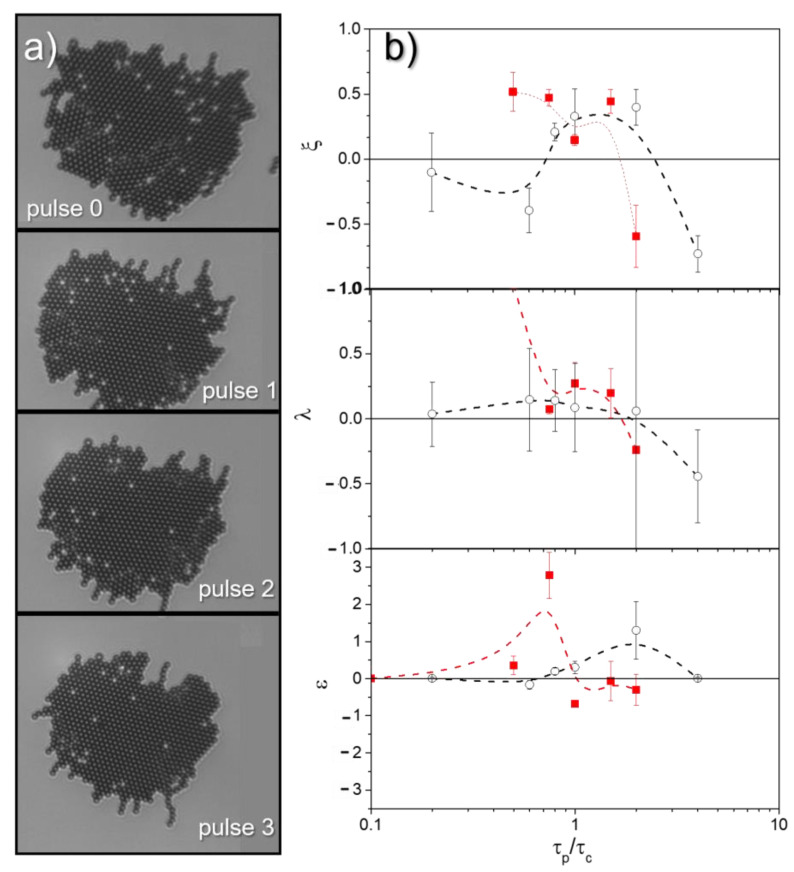
(**a**) The sequence of microscope images shows how the application of a consecutive series of field pulses oriented perpendicular to the fluid interface, where particles are adsorbed, allows the melting of grain boundaries and defects, thus improving crystallinity. (**b**) In small polycrystals composed of between 100 and 500 particles, the pulse application can alter the values taken by the parameters *ξ, α* and *ε*, which evaluate the degree of enhancement of the hexagonal order and are defined throughout the text. These parameters take positive values when the pulse duration is close to the time it takes for the outer particles to travel a radius of distance under the action of the pulses, *τ_c_*. ${µ}_{0}H_{total}^{ext}={µ}_{0}\sqrt{H_{0}^{2}+{\left(H_{z}^{max}\right)}^{2}}=$ 2.0 and 8.0 mT for the black circles and the red squares, respectively.

**Figure 4 nanomaterials-13-00397-f004:**
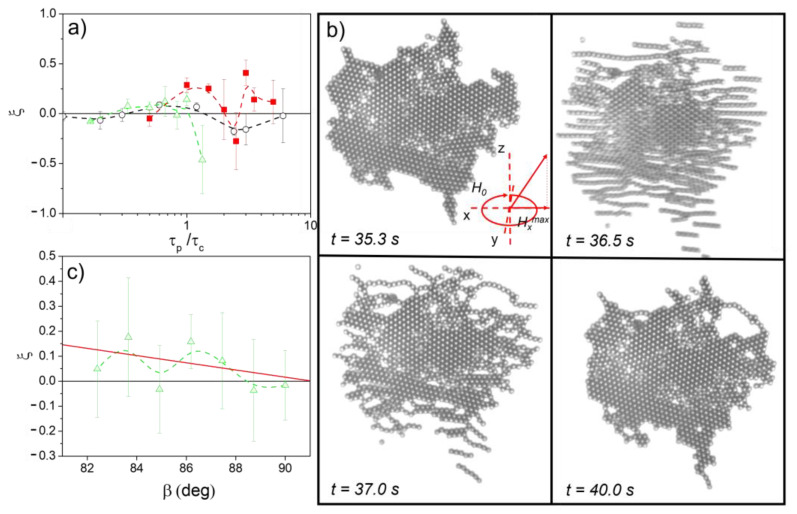
(**a**) In big polycrystals composed of more than 500 particles, the application of field pulses oriented perpendicularly to the fluid interface barely alter the values of ξ. (**b**) A sequence of microscope images showing how the application of tilted pulses, resulting from the simultaneous application of pulses along the Z and X axes, favors the growth of domains oriented along the X direction. (**c**) As β decreases, there is a slight enhancement in the hexagonal order, due to the formation of the privileged direction in the polycrystal. ${µ}_{0}H_{total}^{ext}={µ}_{0}\sqrt{H_{0}^{2}+{\left(H_{z}^{max}\right)}^{2}}=$ 2.0, 5.5 and 8.0 mT for the black circles, green triangles and red squares, respectively.

**Figure 5 nanomaterials-13-00397-f005:**
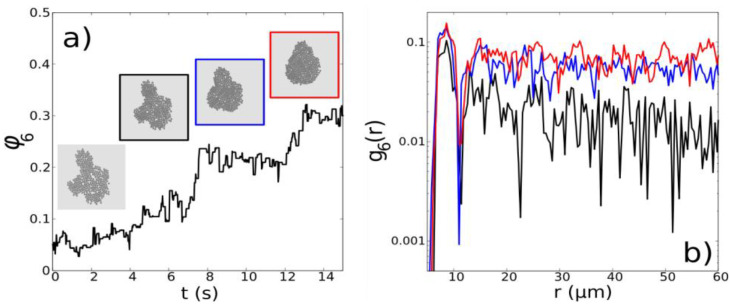
(**a**) Time evolution of $\varphi_{6}\left(t\right)$ along the Brownian dynamics simulation, during the application of three different pulses, alongside snapshots representing the system at the beginning of the simulations, at 4 s (black), 9 s (blue) and 14 s (red). (**b**) Hexagonal order correlation function for the following three different times: 4 s (black), 9 s (blue) and 14 s (red) that correspond with the snapshots of the system shown in panel (**a**). The simulation has been run for $\tau_{p}=0.15\;s$, $\mu_{0}H_{total}^{ext}=5.5\;\mathrm{mT}$, $\mu_{0}H_{0}=3\;\mathrm{mT}$, $T=2\pi /\omega_{pulse}=5\;s$, $f_{x}=20\;\mathrm{Hz}$ and $f_{y}=60\;\mathrm{Hz}$.

## Data Availability

The data that support the findings of this study are available from the corresponding author upon reasonable request.

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
