# Peer review of "Field-Pulse-Induced Annealing of 2D Colloidal Polycrystals"

_nanomaterials, 2023, doi:10.3390/nano13030397_

Round 1
Reviewer 1 Report
The authors have considered magnetic colloids adsorbed on a fluid interface as a model system to investigate the annealing cycles in 2D. The paper seems to be interesting; however, the following comments should be addressed before the publication of the paper:
· Authors need to improve language of the paper. In addition, authors must highlight the scope of the study.
· Abstract must provide precise information of the key findings of the present work. I suggest revising the abstract to highlight the novelty of the work.
· Introduction: I suggest the authors expand the introduction, to include recent studies and to provide enough background and explanations to make the paper informative for a wider researcher.
· The theoretical model of the problem should be explained in details and should be validated by the experimental findings or by the earlier published articles.
· From where the Equation (1) has been considered, there is no reference no calculations?
· It is also suggested to include the physical description of all figures reported in the results and discussion section so that valid conclusions can be drawn from the present work.
· The conclusion section also needs to be improved based on the revisions made in the results and discussions section.
Author Response
We thank all Reviewers for their thorough reading of our manuscript and constructive criticisms. Our point-by-point responses and the corresponding changes made to the manuscript are provided below.
The authors have considered magnetic colloids adsorbed on a fluid interface as a model system to investigate the annealing cycles in 2D. The paper seems to be interesting.
We thank the Reviewer for the positive comment on our manuscript.
However, the following comments should be addressed before the publication of the paper:
Authors need to improve language of the paper.
Following the referee’s suggestions, we have carefully checked the language throughout the paper to improve the general quality of the paper
In addition, authors must highlight the scope of the study.
To highlight the scope of the study, we have included a new first paragraph and a new sentence in the Introduction section, Page 3, line 120:
“Next, we study in detail the effect of polycrystal size, pulse duration, pulse slope and pulse frequency on the efficiency of the annealing process.”
Besides, the objective has been more explicitly highlighted in the new version of the Abstract, as shown in the following point.
Abstract must provide precise information of the key findings of the present work. I suggest revising the abstract to highlight the novelty of the work.
In the new version of the Abstract, we have tried to set out more precisely the scope of the study, together with the main conclusions and the novelty of the work.
“Two-dimensional colloidal crystals are of considerable fundamental and practical importance. However, their quality is often low, due to the widespread presence of domain walls and defects. In this work, we explored the annealing process undergone by monolayers of superparamagnetic colloids adsorbed onto fluid interfaces in the presence of magnetic field pulses. These systems present the extraordinary peculiarity that both the extent and the character of interparticle inter-actions can be adjusted at will, by simply varying the strength and orientation of the applied field, so that the application of field pulses results in a sudden input of energy. Specifically, we have studied the effect of polycrystal size, pulse duration, slope and frequency on the efficiency of the annealing process, and found that (i) this strategy is only effective when the polycrystal consists of less than approximately 10 domains; (ii) that the pulse duration should be of the order of magnitude of the time required for the outer particles to travel one diameter during the heating step, and (iii) that the quality of larger polycrystals can be slightly improved by applying tilted pulses. The experimental results were corroborated by Brownian dynamics simulations.”
Introduction: I suggest the authors expand the introduction, to include recent studies
To the best of our knowledge, the most recent articles on colloidal annealing were already included in the previous version of the manuscript. See, for example, references:
- D. M. Lobmeyer and S. L. Biswal, Science Advances 8, eabn5715 (2022).
- J. L. Dosset, Practical Heat Treating: Basic Principles (Asm International, 2021).
- J. Zhang, J. Yang, Y. Zhang, and M. A. Bevan, Science Advances 6, eabd6716 (2020).
- P.-K. Kao, B. J. VanSaders, S. C. Glotzer, and M. J. Solomon, Scientific Reports 11, 11042 (2021).
- F. Martínez-Pedrero, A. González-Banciella, A. Camino, A. Mateos-Maroto, F. Ortega, R. G. Rubio, I. Pagonabarraga, and C. Calero, Small 17, 2101188 (2021)
- J. Martín-Roca, M. Jiménez, F. Ortega, C. Calero, C. Valeriani, R. G. Rubio, and F. Martínez-Pedrero, Journal of Colloid and Interface Science 614, 378 (2022).
and to provide enough background and explanations to make the paper informative for a wider researcher.
To make the work more accessible to a larger number of researchers, we have started the manuscript with a new paragraph:
“Well-controlled growth of two- and three-dimensional colloidal patterns is essential in the development of photonic crystals, with different applications in electronics, sensors and microlenses [1]. Using an external force to trigger and drive the crystallization process, as well as to promote annealing and tempering seems to be a logical route towards enhancing the quality of microparticle-based nanocrystals [1-3]. In materials science, annealing involves a heating-cooling cycle in which heat rates might be slow and carefully controlled, whereas quenching entails heating the sample to a precise temperature below the critical point. In this work, the structures under study are particle clusters trapped at the interface formed by superparamagnetic particles due to the application of a magnetic rotating field in the plane of the interface [4], while crystalline order enhancement occurs after the application of out-of-plane field pulses which, analogous to what occurs at the atomic level with increasing temperature, can promote the breaking of interparticle bonds and an increase in particle mobility. The objective of the study is to explore the parameter space of the field able to promote order enhancement in colloidal planar polycrystals.”
The theoretical model of the problem should be explained in details and should be validated by the experimental findings or by the earlier published articles.
Since we have not proposed any theoretical model, we assume that the referee means simulations results. In this regard, we emphasize here that section 2 and Figure 5 are devoted to both describe the results of the simulation and to compare the predictions with the experimental results. Moreover, to the best of our knowledge, these results constitute the first attempts to explore annealing process in 2D magnetic nanocrystals under pulsed magnetic field.
From where the Equation (1) has been considered, there is no reference no calculations?
Equation 1 is only a mathematical description of the series of field pulses used in the experiments, which are applied in the direction perpendicular to the plane of the confining interface.
It is also suggested to include the physical description of all figures reported in the results and discussion section so that valid conclusions can be drawn from the present work.
In our humble opinion, Figures 1 and 4 are amply described, both in the corresponding figure captions and throughout the text.
To better explain Figure 2, we have included a new sentence in Page 7, line 320:
“In summary, Figure 2d shows the time evolution of φ6 (t), ξi and the fraction of area covered by the predominant domain, all magnitudes required to calculate the different order parameters ξ, α and ε.”
For the description of Figure 3, we have included a new sentence in Page 9, line 409:
“Figure 3b shows the dependence of the order parameters ξ, α and ε with the duration of the pulses .”
In the description of Figure 5, we have included the sentence, Page 12, line 488:
“In this experiment, three different pulses of duration were applied every 5 seconds. Consequently, increases from 0.1 to 0.3 while the function decays more slowly with distance.”
The conclusion section also needs to be improved based on the revisions made in the results and discussions section.
We have included new sentences in the Discussion section that tries to better describe the experiments and results obtained in this work:
Page 13, line 518: “We have investigated the dependence of different order parameters, which reflect the degree of crystallinity of the studied polycrystals, as a function of their size as well as the main characteristics of the applied pulses (duration time, frequency, intensity, inclination).”
Page 13, line 528: “an ordered array of interstitial voids that act as a diffraction grating when the interstitial spacing is similar to the wavelength of the incident light”
Reviewer 2 Report
It is an interesting paper on filed-pulse induced annealing of 2D colloidal polycrystals. The authors introduce an annealing strategy based on the application of magnetic field pulses. The experimental results were colloborated by Browninan dynamics simulations.
The paper is well organized, and I would like to recommend the publication of the paper after minor changes.
1. It is required to discuss the role of magnetic dipole-dipole interaction on the annealing of 2D colloidal polycrystals. Furthermore, it is required to discuss whether mesocrystal can be formed or not in the manuscript. See J. Phys. Chem. C 116, 319 (2012), 119, 24597 (2015).
Author Response
We thank all Reviewers for their thorough reading of our manuscript and constructive criticisms. Our point-by-point responses and the corresponding changes made to the manuscript are provided below.
It is an interesting paper on filed-pulse induced annealing of 2D colloidal polycrystals. The authors introduce an annealing strategy based on the application of magnetic field pulses. The experimental results were colloborated by Browninan dynamics simulations.
The paper is well organized, and I would like to recommend the publication of the paper after minor changes.
We thank the reviewer for all the positive comments on our manuscript.
It is required to discuss the role of magnetic dipole-dipole interaction on the annealing of 2D colloidal polycrystals.
In the new version of the Introduction, we have included a brief discussion about the role of magnetic interactions, Page 3, line 110:
“The use of magnetic colloids in the study of the annealing of 2D colloidal polycrystals is justified since they allow the induction of interparticle interactions with different characteristics -range, anisotropy, and intensity-, easily tuneable by the action of an externally applied magnetic field of moderate intensity. In the explored configuration, in which the particles are forced to reside in the plane of the interface,…”
And Page 6, line 262: “In the dipole-dipole approximation, the time-averaged potential of two paramagnetic spheres adsorbed onto a planar fluid interface when under the action of a field in precession about the Z axis can be expressed as the sum of the in- and out of plane contributions [37], 〈U_dd 〉=(μ_0 χ^2)/(2πr^3 )(〖〖〖(H〗_z^max)〗^2/2-H〗_0^2).”
Additional comments regarding the dipole-dipole interactions were introduced in Page 4, line 188: “At low frequency, this time dependent field forces the formation of elongated structures that vibrate or rotate, synchronously or asynchronously, with the applied field [43]. If the frequency of the field is higher than 10 Hz, however, an averaged attractive potential promotes the formation of circular assemblies or chains composed of rotating particles which in turn rotate or vibrate in the plane of the field [37,44]. When Hx and Hy are similar, the application of the high frequency rotating field induces an isotropic effective attractive dipolar potential at the interface plane, 〈U_dd 〉=-[(μ_0 m^2)⁄(8π〖(x^2+y^2)〗^(3/2) )], where μ0 is the vacuum magnetic permeability and m the induced magnetic moment in the particles, and promotes the formation of two-dimensional colloidal polycrystals.”
Furthermore, it is required to discuss whether mesocrystal can be formed or not in the manuscript. See J. Phys. Chem. C 116, 319 (2012), 119, 24597 (2015).
In connection with this point, we have included a new sentence in the Discussion Section, Page 13, line 532:
“In addition, analogous methods could be used in the controlled production of 2D mesocrystals, aggregates of nanocrystals with aligned crystalline axes, when the latter have magnetic properties and relatively large sizes [55].”
Together with the new reference 55: K. Yasui and K. Kato, The Journal of Physical Chemistry C 119, 24597 (2015)